Establishing an ecological security pattern for urban agglomeration, taking ecosystem services and human interference factors into consideration

http://orcid.org/0000-0001-7422-4291 Wang Dongchuan 1 2 mrwangdc@126.com
Chen Junhe 1
Zhang Lihui 1
Sun Zhichao 1
Wang Xiao 1
Zhang Xian 1
Zhang Wei 1
1 School of Geology and Geomatics, Tianjin Chengjian University , Tianjin , China
2 Tianjin Key Laboratory of Civil Structure Protection and Reinforcement , Tianjin , China
Marino Bruno
Electronic publication date: 2019 Jul 15
Publication date: 2019
Volume: 7
Electronic Location ID: e7306
Received 2019 Mar 8; Accepted 2019 Jun 17
Copyright: © 2019 Wang et al.
Copyright year: 2019
Copyright holder: Wang et al.
License: This is an open access article distributed under the terms of the Creative Commons Attribution License, which permits unrestricted use, distribution, reproduction and adaptation in any medium and for any purpose provided that it is properly attributed. For attribution, the original author(s), title, publication source (PeerJ) and either DOI or URL of the article must be cited.
License URL: https://creativecommons.org/licenses/by/4.0/

Keywords: Beijing–Tianjin–Hebei urban agglomeration, Ecosystem service, Ecological security pattern, Ecological source, Ecological corridor

Funding: Major project of the National Natural Science Foundation of China 41590841 National Key Research and Development Plan 2016YFC0503001 Natural Science Foundation of Tianjin, China 18JCYBJC90900 Project of Scientific Research Plan of Tianjin Education Commission 2018KJ164 This research is funded by the Major project of the National Natural Science Foundation of China (No. 41590841), the National Key Research and Development Plan (No. 2016YFC0503001), the Natural Science Foundation of Tianjin, China (No. 18JCYBJC90900), and the Project of Scientific Research Plan of Tianjin Education Commission (No. 2018KJ164). The funders had no role in study design, data collection and analysis, decision to publish, or preparation of the manuscript.

==============================
The assessment of ecological security patterns is a topic of conversation in landscape ecology in recent years. However, ecosystem services and human activities are seldom considered comprehensively in the assessment of ecological security patterns. The present study employs the Beijing–Tianjin–Hebei urban agglomeration as a study area, and uses ecological services to determine the ecological sources. The importance of ecological sources is classified based on logical coding and functional types of ecological services. The research combines regional characteristics to select and quantitatively calculate three human disturbance factors: soil erosion sensitivity, geological hazard sensitivity, and night lighting. Then the basic surface resistance of land use to limit migration is modified and ecological corridors are identified by combining these three disturbance factors. The results indicate that the sources of water production, soil and water conservation, and carbon fixation are mainly provided in mountainous areas, recreation sources are mostly distributed in the plains, and these ecological sources improve the maintenance of ecological corridors. The modification of resistance surfaces significantly changes the length of ecological corridors in Tianjin, Tangshan, Cangzhou, and Beijing, and the modified resistance surface improves the recognition of ecological corridors. This study provides a new research framework for identifying the ecological security patterns of urban agglomerations and provides scientific guidance related to ecological protection and urban planning for the Beijing−Tianjin−Hebei urban agglomeration.

Introduction

While urbanization provides beneficial results to modern civilization, it also creates a series of ecological and environmental problems, such as the loss of natural landscapes, a decline in ecosystem services, and the aggravation of environmental health risks, which can affect the sustainable development of cities (Han, Liu & Wang, 2015). Therefore, ensuring the stability and functioning of urban ecosystems while promoting sustainable urban development has become a major issue facing the international community (Cumming & Allen, 2017).

In general, the goal of establishing an ecological security pattern is to achieve regional ecological sustainability through integrating landscape patterns with ecological processes while comparing the importance of different landscape patches to specific ecological processes and ecosystem services (Peng et al., 2018d). Documenting ecological security patterns can restrict the expansion of urbanization, regulate ecological processes reasonably, maintain material and energy cycles, strengthen urban ecological health, and promote sustainable urban development (Li et al., 2011).

Since the 1990s, many scholars have conducted extensive research on the development of ecological security patterns, including theoretical exploration, index development, and method realization, they have achieved a series of important results (Ahern, 1995; Zube, 1995; Albanese & Haukos, 2017; Miao et al., 2015). The scope of ecological security pattern research includes species conservation (Dalang & Hersperger, 2012), land management (Gaaff & Reinhard, 2012), urban planning (Dong et al., 2015), and habitat protection (Kang et al., 2016). The assessment of ecological security patterns is a topic of conversation in landscape ecology in recent years. However, ecosystem services and human activities are seldom considered comprehensively in the assessment of ecological security patterns (Peng et al., 2018d; Zhang et al., 2017; Jing, Chen & Sun, 2018; Dong et al., 2015). Currently, the development of ecological security patterns utilizing ecosystem supply needs has formed a research paradigm including the identification of ecological sources, and ecological corridors (Klar et al., 2012; Gu et al., 2016; Yang et al., 2018). The first step in ecological security pattern development is to identify ecological sources, mainly combining large-scale habitat patches, nature reserves and scenic spots to directly select the ecological source (Teng et al., 2011; Gurrutxaga, Lozano & Barrio, 2010), or through ecological sensitivity, ecological importance, landscape connectivity, and evaluation of ecological suitability (Su et al., 2016; Kong et al., 2010; Zhang et al., 2017). Ecological sources are the areas that provide the individuals involved in species dispersal, maintenance, and landscape components to promote the development of ecological processes (Chen, Fu & Zhao, 2008). Ecological sources are identified by quantitative assessment of regional ecological security patterns (Wu et al., 2013). Ecosystem services refers to the environmental conditions and effects on which human beings depend for survival and development. These services include not only food, fresh water, and raw materials for industrial and agricultural production provided by ecosystems but more importantly, these services support and maintain of ecosystems (Daily, 1997). Therefore, it is of great significance to identify the areas that provide the highest level of ecological service as the ecological source areas for maintaining urban ecological security (Peng et al., 2018d). However, most of the current research identified ecological source areas by considering the function of ecological services, they did not consider the effects of different types of ecological services on the importance of the same ecological source. The definition and functional classification of ecological sources is not only the basis for the formulation of relevant ecological policies, but also the former must be clear for urban development and construction. The problem is related to the rational use of land resources and the sustainable development of human society (Egoh & Reyers, 2008). In recent years, most studies have shown that clarifying the impact of ecological service categories on the importance of the ecological source areas is helpful in understanding the corresponding relationship between ecological source areas and ecological service functions. Moreover, understanding the relationships among multiple ecosystem services and the mechanisms behind these relationships will improve our ability to sustainably manage landscapes in order to provide multiple ecosystem services (Bennett, Peterson & Gordon, 2009) toward an integrated ecological sources management (Raudsepp-Hearne, Peterson & Bennett, 2010; Kareiva et al., 2007).

The next step in ecological security pattern development is to identify ecological corridors. In the extraction of ecological corridors, the most commonly used method is the GIS-based modeling approach proposed by Knaapen, Scheffer & Harms (1992) and improved by Yu (1999) based on the minimum cumulative resistance (MCR) model. GIS is a comprehensive technology for processing geospatial information. It can not only express visible spatial positional relationships, but also reflect spatial locations or related numerical values, words and charts that reflect the characteristics of things (Beni et al., 2011). Urban ecological planning is necessary to accurately judge and decide the spatial layout relationship between urban ecological environment and urban development and construction, and as a complex and comprehensive research field, it needs the abstract data of various spatial information to be processed (Aretano et al., 2015). GIS as a decision support tool for environmental protection and nature reserve protection can play a targeted role in spatial relationship processing and data analysis. The GIS-based modeling approach has been used for numbers of urban planning applications in China (Su et al., 2016; Dong et al., 2015), where resistance is the degree of impediment to species migration between landscape units. Currently, the development of resistance surfaces is generally based on developing a resistance coefficient related to ecological factors such as land cover type and slope (Fu et al., 2010). The method does not consider the spatial differences caused by specific ecological problems in different regions. The correction of the resistance surface is also based on nighttime lighting (Zhang et al., 2017) or imperious index data (Peng et al., 2018c) that are used to enhance the spatial differences in resistance surfaces. The method considers the spatial differences caused by specific ecological problems in resistance surfaces, but is generally based on the qualitative evaluation of ecological security patterns for a single problem, the extent of the spatial coverage is generally aimed at the province (Peng et al., 2018a), city (Li et al., 2010), or county level (Yu et al., 2018). As a relatively complex ecosystem, an urban agglomeration involved various aspects of ecological security, each of which included many factors influencing urban conditions. A comprehensive evaluation cannot be done by studying ecological security based on a single factor (Chen, Jing & Sun, 2018).

The health and sustainable development of the Beijing–Tianjin–Hebei urban agglomeration is important ensurancing of china’s national security. However, human activities become more frequent with intense urbanization, which leads to a series of ecological problems (Gong et al., 2009; Liu et al., 2017; Duan et al., 2015). Among them, soil and water conservation serve as an important ecological safeguards in support of economic and social development these should play an important role in the coordinated development of the Beijing–Tianjin–Hebei region (Chen et al., 2017; Miao et al., 2011). In addition, the fragile ecological base, changing climatic conditions, and intense activity frequently cause geological disasters in the Beijing–Tianjin–Hebei urban agglomeration (Meng et al., 2017). Soil erosion and geological hazard sensitivity indices reflect the degree of ecosystem response to human disturbance and natural environment changes (Ouyang, Wang & Miao, 2000). Night light data are widely used in the study of population density, economic development, and urban heat island effects, etc. These spatial measures—indicators are a comprehensive characterization of the intensity of human activity in an area (Mellander et al., 2015). In areas that are highly sensitive to soil erosion and geological hazards or that have nighttime light, the intensity of human activity is normally relatively high, which has a certain effect on regional the migration of urban species. Therefore, in setting the Beijing–Tianjin–Hebei urban agglomeration as the study area, this study employs an analysis of ecosystem services and human disturbance factors to construct ecological security patterns, with the goal of reducing the negative effects of human activities on the ecological security pattern of the Beijing–Tianjin–Hebei urban agglomeration. Hence, the research objectives were proposed as follows: (1) One objective was to determine the ecological source of various resources through a quantitative evaluation of water production services, soil and water conservation, carbon fixation services, as well as leisure and recreation services; (2) to determine the importance of dividing ecological sources by combining logical coding and ecological service function types; finally, (3) using the soil erosion sensitivity, geological hazard sensitivity, and nighttime light indices, we wanted to modify ecological resistance surfaces based on land use to identify ecological corridors.

Methodology

Ecological security patterns combine landscape pattern with ecological processes to achieve regional ecological sustainability by comparing the importance of different landscape patterns (Peng et al., 2018d). The ecological security pattern of the Beijing–Tianjin–Hebei urban agglomeration constructed in this study consists of four steps: firstly, ecological services are used to determine the ecological sources. Second, the importance of the ecological sources is determined by combining the logical coding and identification of the type of ecological services. Thirdly, the basic resistance surface is modified with human interference factors. Finally, the ecological corridor is extracted by using the MCR model. The flow chart of this study is shown in Fig. 1.

Figure 1 Overview of the construction of ecological security pattern.

Flow chart of ecological security pattern construction. Arrows indicate the relationship between the various elements.

Study area and data sources

The Beijing–Tianjin–Hebei urban agglomeration lies in the coastal area of Eastern China, and covers a total area of 218,000 km2. In this region, the Yanshan Mountains delimit the north, the North China Plain lies to the south, the Taihang Mountains stand to the west, and the Bohai Sea borders the eastern edge of the region (Fig. 2). The cold and snowy climate in winter gives way to an arid spring with high wind speeds and frequent sandstorms, summers are hot and rainy, which is conducive to rock and soil weathering. According to a survey, by 2015, a total of 7,255 geological hazards and hidden dangers were found in the study area, including 3,526 collapses, 770 landslides, 2,288 debris flows, and 671 ground collapses (Meng et al., 2017). The human population accounts for 7.8% of the total population of China, making this urban agglomeration a densely populated area. Data preprocessing and data processing are mainly done with GIS tools. The data sources are shown in Table 1.

Figure 2 Location of the Beijing–Tianjin–Hebei urban agglomeration and digital elevation model (DEM) of the study area.

Table 1 Data sources.

Data	Remarks	Data sources	
Meteorological data	It was derived from the daily dataset of basic meteorological elements from the China Meteorological Association managers of China’s national surface meteorological stations (V3.0). Data preprocessing programs mainly included Kriging interpolation and cropping, etc.	http://data.cma.cn/	
Land use	They were obtained by visual interpretation of Landsat8 OLI imagery. Date range: July 12, 2015, July 28, 2015. Data preprocessing procedures mainly included atmospheric correction, radiometric calibration, and band fusion.	http://www.gscloud.cn/	
Digital elevation model	Data acquisition from a geospatial data cloud with spatial resolution of 30 m. Data preprocessing programs mainly included cropping, splicing and reprojection.	
Soil texture data	Data acquisition from the Soil Science Data Center of the National Earth System Science Data Sharing Service Platform. Data preprocessing programs mainly included vector to raster, reprojection, etc.	http://www.geodata.cn/data/	
Soil depth data	Data acquisition from the Science Data Center of Cold and Dry Areas. Data preprocessing programs mainly included vector to raster, reprojection, etc.	http://westdc.westgis.ac.cn/	
Normalized difference vegetation index	They were derived from the publishing system of global change scientific research data. Data preprocessing programs mainly included geometric correction, visual interpretation, and re-projection (Kuang, Yang & Yan, 2017; Gao et al., 2017)	http://www.geodoi.ac.cn/WebCn/doi.aspx?Id=959	
Net primary production	http://www.geodoi.ac.cn/WebCn/doi.aspx?Id=215	
Geological hazard survey data	Geological hazard survey data were based on geological hazard general survey data at a scale of 1:100,000 and detailed survey data at a scale of 1:50,000 for geological hazards, as well as geological hazard survey data such as the karst collapse database.	http://www.mnr.gov.cn/	
Beijing ecological protection red line	The red line of ecological protection refers to a strict land use control boundary demarcated by Chinese law in important ecological functional, ecological sensitive, and vulnerable areas, and represent the last line of defense for national and regional ecological security. Data preprocessing programs mainly included geometric correction, visual interpretation, and re-projection.	http://www.bjepb.gov.cn/bjhrb/xxgk/fgwj/qtwj/tzgg/834706/index.html	
Tianjin ecological protection red line	http://www.h2o-china.com/news/280469.html	
Hebei ecological protection red line	http://www.myzaker.com/article/5b39d9ca1bc8e0bf3a00022c	

Ecological sources

Ecological service function assessment

Assessment of the importance of ecosystem services frequently serves as a basic method used to identify regional sources of ecological services. This is done by analyzing variation in regional ecological services and identifying the areas that are most important for the maintenance of typical regional ecosystems. Therefore, the present study quantitatively identifies and evaluates services of water production, water and soil conservation, carbon fixation, and those related to leisure and recreation of the urban agglomeration in combination with an analysis of the characteristics of the natural environment of the Beijing–Tianjin–Hebei urban agglomeration.

In particular, the function of water production is based on the water cycle. Water production depends on the rainfall in a grid unit minus the actual evapotranspiration (Miao, Ni & Borthwick, 2010). In the present study, the Integrated Valuation of Ecosystem Services and Trade-offs water yield assessment model was used to evaluate the ecological service functions in this region. The Integrated Valuation of Ecosystem Services and Trade-offs model is a tool used for the comprehensive assessment and balancing of ecosystem services developed jointly by Stanford University, the Worldwide Fund for Nature, and the Nature Conservancy.

The revised universal soil loss equation (USLE) model was used to evaluate soil and water conservation in the studied region. The USLE model, namely the general soil loss equation, was originally established by Wischmeier & Smith (1965) based on a large amount of plot observation and simulated rainfall experimental data. In 1992, The United States Department of Agriculture-Agricultural Research Service improved the USLE model and initially proposed the revised USLE model for soil erosion analysis. The formula is provided in Eq. (1): (1) Ac=Ap−Ar=R×K×L×S×(1−C)

where Ac is the amount of soil and water conservation, t/(hm2·a); Ap is the potential amount of soil erosion; Ar is the actual amount of soil erosion; R is the factor of rainfall erosivity, MJ·mm/(hm2·h·a); K is the factor of soil erodibility, t·hm2·h/(hm2·MJ·mm); L and S are the topographic factors; L are the factor of slope length. S is the slope factor, and C is vegetation cover management factor.

The service of carbon fixation was evaluated basing on the principle of light energy utilization. The estimation formula is shown in Eq. (2): (2) NPP=PAR×FPAR×ε*×f(w)×f(t)−Ra

Where PAR is photosynthetically active radiation, the unit is MJ/m2; FPAR is the absorption component of the vegetation layer to the incident photosynthetically active radiation, no unit; (ε* is the maximum light energy utilization rate, the unit is gC/MJ; f(t) is the influence factor of temperature on the utilization of light energy, no unit; f(w) is the influence factor of water on the utilization of light energy, no unit; Ra is the autotrophic respiration of vegetation, the unit is gC/m2.

The leisure and recreation service mainly considers the wetlands of the Beijing–Tianjin–Hebei urban agglomeration. Buffer zones of pedestrian walking distance to these wetlands of 5, 10, 15, 20, and 25 min were analyzed, which are included in different levels of leisure and recreation, aiming at emphasizing the effects of human activities (Peng et al., 2018b).

Ecological source level

Based on quantitatively evaluating the ecosystem services of the studied region and identifying the ecological source areas, logical coding is introduced to identify the ecosystem services of different ecological sources. This coding is used to represent one or more types of ecosystem services contained in the ecological source area. The logical coding in this study uses “1” or “0” to indicate that a grid has or does not have a particular ecosystem service, respectively. In the present study, the raster computing function of ArcGIS ver. 10.2 was combined to obtain the spatial pattern of the logical coding of each ecological service category. Equation (3) provides the logical coding system: (3) Tij=(G1)ij×10n−1+(G2)ij×10n−2+…+(Gn)ij×10n−n

in which T is the logical coding value of column j and row i in the raster image of logical analysis results, which indicates one or more ecosystem service categories, and n is the number of ecosystem services. The present study looked at four ecosystem services. Therefore, n = 4 and G1ij, G2ij, G3ij, and G4ij are the logical coding of the corresponding raster types on the raster images of the services of water production, water and soil conservation, carbon fixation, and leisure with recreation, respectively. For example, if the type logical coding is “1101,” it means that the ecological source area has the services of water production, water and soil conservation and leisure, and does not have the service of carbon fixation.

In the present study, the ecological source areas were classified as described above based on the logical coding to determine the importance of each ecological source area. If three or four of the ecological services are coded using the number “1,” the area is determined to be a primary ecological source; if only two or one are coded using the number of “1,” the area is determined to be a secondary or tertiary ecological source.

Resistance surface

The concept of an ecological resistance surface refers to the degree of impediment to species migration between landscape units. The spatial differences of specific ecological problems in small-scale areas can often be solved by constructing desirable ecological resistance surface based on the single-factor correction of land cover types. However, as a relatively large urban agglomeration, land managers of the Beijing–Tianjin–Hebei region should not only consider the ecological problems confined to specific areas, but should also analyze the region as a whole. Geological conditions in the Beijing–Tianjin–Hebei urban agglomeration are complex and fragile, landslides, debris flows, and serious soil erosion occur frequently in the region. Increasingly human activities interfere with the natural landscape conditions, this is especially true as it relates to the expansion of urbanization. These factors will have a great impact on species migration and regional ecological security and stability. Therefore, in this study, sensitivity indices tied to geological hazards, soil erosion, and night light data are introduced to modify the basic ecological resistance surface based on land use. In addition, the ecological red line corridors were manually vectorized in the Beijing, Tianjin, and Hebei Red Lines of Ecological Protection in 2018. Initially, ecological resistance is assigned to 0 and finally superimposed on the basic resistance surface, then the revised resistance surface is converted into raster format. Among, the above three sensitivity indices, that of soil erosion adopts the evaluation indexes of precipitation erosivity, soil erodibility, slope length, and surface vegetation cover, which have been proposed based on the red line technical guidelines for ecological protection of Yang et al. (2016). The sensitivity index of geological hazards used nine indicators to calculate the total distribution weight of the layers (Table 2). The weight of each index was determined by analytic hierarchy process (AHP) (Saaty, 1987). The AHP is completed by Yaahp software, which is a comprehensive evaluation assistant software based on the AHP and fuzzy comprehensive evaluation method. The consistency test of the results of the AHP is carried out where the consistency index value is 0.1056, because the calculation of geological hazard sensitivity includes nine indicators, namely n = 9, and the table is found, random index = 1.45, consistency ratio = 0.0728 < 0.1, indicating the judgment matrix has satisfactory consistency. In this paper, with the sensitivity analysis in Yaahp software, the number of sampling points is set to 100, and the corresponding total ranking weights when the middle layer elements change from 0 to 1 are calculated. The basic sensitivity indicators of disaster, terrain and other are 0.0423, 0.0317, and 0.0255, respectively. This indicator highlights the maximum range of weight change of each option caused by the change of the current element weight. Finally, the three factors were normalized and replaced by the specific ecological resistance surface correction as follows in Eq. (4): (4) Ri=(wb×NLiNLa+wc×SEiSEa+wd×GSiGSa)×R 

Table 2 Weight of various factors of geological hazard sensitivity.

Decision objective	Interlayer element	Alternative	Weight	
Geological hazard sensitivity	Disaster	Density of geological hazards	0.1908	
Euclidean distance of faults	0.1349	
Earthquake acceleration	0.1099	
Terrain	Slope	0.156	
Waviness	0.0912	
Slope position	0.0632	
Other	Engineering rock group	0.1109	
Normalized difference vegetation index	0.0743	
Precipitation	0.0689	

where Ri is the ecological resistance coefficient of grid i based on the water loss sensitivity, night lighting, and geological hazard sensitivity indices, NLi is the night light of grid i, NLa is the average night light of the land use type a corresponding to grid i; SEi is the water and soil erosion sensitivity index of grid i; SEa is the average soil erosion sensitivity index of land use type a corresponding to grid i. GSi is the sensitivity index of geological hazards of grid i and GSa is the average geohazard sensitivity index of land use type a corresponding to grid i, and R is the basic resistance coefficient of land use type corresponding to grid i. Since it is impossible to determine which factors have a greater impact on the resistance surface of an urban agglomeration, the weight of night lighting, soil and water loss sensitivity index, and geological hazard sensitivity index is set to 1:1:1, that is wb:wc:wd = 1:1:1.

Ecological corridors

Constructing ecological corridors can solve the problem of fragmentation related to ecological sources and enhance the connectivity among ecological sources. With the rapid development of the Beijing–Tianjin–Hebei urban agglomeration, the islanding and fragmentation of the local landscape has increased dramatically. The MCR model was first proposed by Knaapen, Scheffer & Harms (1992). This model considers three factors, source, distance, and landscape interface, to calculate the cost to a species caused by moving from a source to a destination, and it extracts the resistance trough between two adjacent “sources” and the most easily connected low-resistance channel as a corridor between ecological safe habitat sources as shown in Eq. (5): (5) MCR=fmin∑j=ni=mDij×Ri

where MCR is the MCR value; Dij is the spatial distance of species from source J to landscape unit i, Ri is the resistance coefficient of landscape unit i; and f is the positive correlation between MCR and ecological process.

Results

Spatial pattern of ecological sources

Spatial pattern of ecological services

The spatial pattern of a single ecosystem services can reflect the effects of different ecological processes on regional ecological security. The assessment results of ecosystem services were divided into five levels from low to high by the natural breakpoint method as follows: generally, unimportant and slightly, moderately, highly, and extremely important (Fig. 3), at the same time, the area and area ratio occupied by important areas of various ecosystem service was calculated (Table 3).

Figure 3 Spatial patterns of ecological services.

Spatial patterns of (A) water production, (B) soil and water conservation, (C) carbon fixation, (D) leisure and recreation services.

Table 3 Area and proportion of different levels of ecosystem services.

	Carbon fixation service	Soil and water conservation service	Water production service	Leisure and recreational service	
Area (km2)	Proportion (%)	Area (km2)	Proportion (%)	Area (km2)	Proportion (%)	Area (km2)	Proportion (%)	
Generally important	30,006.56	13.88	167,863.23	78.31	100,565.55	46.97	4,869.16	2.25	
Slightly important	79,240.88	36.66	11,425.31	5.33	64,397.39	30.08	5,784.58	2.68	
Moderately important	77,111.86	35.67	9,489.43	4.43	17,910.11	8.37	6,724.76	3.11	
Highly important	9,691.46	4.48	7,389.25	3.45	17,691.88	8.26	7,697.28	3.56	
Extremely important	20,117.15	9.31	18,180.17	8.48	13,528.04	6.32	8,701.79	4.03	

In the Beijing–Tianjin–Hebei urban agglomeration the water production service was generally higher in mountainous and hilly areas than that of plain and plateau areas, it is also higher in the western and eastern areas than in the central areas (Fig. 3A; Table 3). Most of the extremely important areas are located west of Handan and Xingtai, northwest of Baoding, and in a few scattered areas of the eastern coastal areas including Qinhuangdao, Tangshan, and Cangzhou. Most of these areas are located in the windward slopes of the Yanshan and Taihang Mountains that have a large amount of topographical relief and receive relatively higher amounts of precipitation, so the overall water production capacity is strong there. The total amount of the moderately and highly important areas for the water production service is not very different, the proportions of these areas were 8.37% and 8.26%, respectively, and they were mainly distributed in the central plain area. Moreover, the spatial extent of generally important areas was the largest, accounted for 46.97%, indicating that this urban agglomeration belongs to an area of serious water shortage.

In the Beijing–Tianjin–Hebei urban agglomeration the distribution of the soil and water conservation of area gradually decreased from the Yanshan and Taihang mountains to the surrounding areas (Fig. 3B; Table 3). The aforementioned area was the largest covering a total amount of 167,863.23 km2, accounting for 78.31% of the total study area. Except for extremely important areas, the proportion of other grades was not very different. The combined areas of the moderately, highly, and extremely important regions was 35,058.85 km2, accounting for only 16.36% of the total area, indicating that the overall soil and water conservation capacity of the region is relatively weak. As an important ecological area of soil and water conservation in the Beijing–Tianjin–Hebei urban agglomeration, these areas are rich in precipitation, while the factors of soil erodibility are small, or the vegetation cover has a good ability to regulate and store precipitation; that is, these areas have a strong ability conserve water and soil.

In the Beijing–Tianjin–Hebei urban agglomeration the carbon fixation service is higher in the north and lower in the South and higher in the west and lower in the east (Fig. 3C; Table 3). The spatial differences among the important regions are relatively large. The areas of the highly and extremely important regions were 9,691.46 km2, and 20,117.15 km2, respectively, with the area ratios of 4.48% and 9.31%, respectively. They were mainly distributed in a small part of Handan, Xingtai, Shijiazhuang, Baoding, and Beijing and most of Chengde. Most of the land use types in these areas were forest, with abundant vegetation coverage, a humid climate, and good hydrothermal conditions, which can better meet plant growth requirements than other habitats. The area of moderately important areas covered 77,111.86 km2 which accounted for 35.67% of the study area. Most of moderately important areas were surrounded by highly and extremely important areas. The areas of generally and slightly important areas were 30,006.56 km2 and 79,240.88 km2 which accounted for 13.88% and 36.66% of the study area, respectively. These areas are basically developed but unused areas that are mainly distributed in the central, southern, and eastern parts of the region.

The leisure and recreational service of the Beijing–Tianjin–Hebei urban agglomeration presented a multi-point distribution pattern as a whole (Fig. 3D; Table 3). This service was mainly distributed in the plateau area of Zhangjiakou, the eastern area of Beijing, the southeastern area of Tianjin, and along the coastal area. The spatial extent of the most important area was 8,701.79 km2, and it accounted for 4.03% of the entire area.

Spatial pattern of the level of ecological sources

The extremely important areas of each ecosystem service were selected as ecological source areas. The spatial pattern of the logical coding of the ecological service categories is shown in Fig. 4. The area and proportion of logical coding of each ecosystem source type was also calculated (Table 4).

Figure 4 Spatial patterns of ecological sources.

Spatial patterns of the (A) logical coding of ecological service categories and (B) levels of ecological sources.

Table 4 Area and proportion of logical coding of ecological service categories.

Logical coding of ecological sources type	The ecological service type corresponding to the logical code of ecological sources	Area of ecological sources type (km2)	Area ratio (%)	
0000	Non-ecological sources	133,585.7	61.79	
0001	Leisure and recreation services	1,978.31	0.92	
0010	Carbon fixation services	10,434	4.83	
0100	Soil and water conservation services	25,477.83	11.78	
0110	Soil and water conservation services, carbon fixation services	20,520.31	9.49	
1000	Water production services	11,372.3	5.26	
1010	Water production services, carbon fixation services	1,354.78	0.63	
1100	Water production services, soil and water conservation services	4,702.49	2.18	
1110	Water production services, soil and water conservation services, carbon fixation services	6,763.19	3.13	

The Beijing–Tianjin–Hebei urban agglomeration had eight types of ecological source areas (Fig. 4A; Table 4). The area of ecological sources covered 82,605.78 km2, accounting for 38.21% of the total area. Among them, the plain area provided the source of leisure and recreation services, and the source logical coding containing this function is 0001, accounting for 0.92% of the area of this urban agglomeration. The mountainous area is the source of water production, soil and water conservation and carbon fixation services. Among the four types of ecological sources, carbon fixation sources accounted for the largest proportion.

Most of the mountain areas are first or second-level ecological sources (Fig. 4B). Plains are typically a third-level ecological source. The ecological source areas were primarily distributed in the regions of dense mountains and sparse plains. The numbers of ecological sources at different levels were counted. A total of 15, 38, and 97 primary, secondary, and tertiary ecological sources areas were identified, respectively. The spatial extent of primary, secondary, and tertiary ecological source areas was 27,999.21 km2, 25,378.32 km2, and 15,505.07 km2, respectively. Mountainous areas provided first and second-level ecological source areas, and the plains provided third level ecological source areas. The results indicate that the habitat quality of the Beijing–Tianjin–Hebei urban agglomeration is better than that of the plains area, and the source land of the plains area is severely fragmented.

Spatial pattern of ecological corridors and resistance surfaces

Resistance surface modification is the core part of the development of an ecological security pattern. The Beijing–Tianjin–Hebei urban agglomeration frequently experiences geological disasters, serious soil erosion, and frequent human activities. Therefore, this study uses soil erosion sensitivity, geological hazard sensitivity, and night lighting to modify the resistance surface directly based on land use patterns. The averages of the soil erosion sensitivity index, nighttime light, geological disaster sensitivity index, and resistance values of each city in the Beijing–Tianjin–Hebei urban agglomeration were calculated individually (Table 5). At the same time, the average indices of soil erosion sensitivity, nighttime light index, and geological hazard sensitivity of various land use types were calculated (Table 6).

Table 5 The average soil erosion sensitivity.

	Average soil erosion sensitivity	Average night light	Average geological hazard sensitivity	Average resistance value	
Chengde	0.17	0.04	0.18	109.13	
Zhangjiakou	0.12	0.05	0.16	200.72	
Qinhuangdao	0.12	0.13	0.17	250.93	
Baoding	0.1	0.13	0.16	255.8	
Shijiazhuang	0.09	0.18	0.15	301.86	
Beijing	0.14	0.33	0.18	311.67	
Xingtai	0.08	0.15	0.15	331.7	
Hengshui	0.05	0.13	0.14	339.74	
Cangzhou	0.05	0.18	0.14	346.57	
Handan	0.07	0.2	0.15	358.21	
Tangshan	0.08	0.25	0.17	370.29	
Tianjian	0.06	0.43	0.16	452.22	
Langfang	0.05	0.34	0.16	458.8	

Table 6 The average soil erosion sensitivity index, average nighttime light index, average geological disaster sensitivity index of various land use.

	Average soil erosion sensitivity	Average night light	Average geological hazard sensitivity	
Forest land	0.17	0.05	0.19	
Wetland	0.04	0.26	0.14	
Grassland	0.11	0.07	0.17	
Farmland	0.07	0.17	0.15	
Unused land	0.08	0.18	0.14	
Construction land	0.06	0.42	0.15	

Higher values for soil erosion sensitivity index (Fig. 5A) were mainly distributed in the Taihang Mountains, the intermountain basins in the northwest portion of the Hebei Province, and the Yanshan Mountains in north China. The average sensitivity index of soil erosion of each land use type was measured independently. The average soil and water loss sensitivity index values of Chengde, Beijing, Qinhuangdao, and Zhangjiakou were relatively large, The average soil and water loss sensitivity index values for Tianjin, Hengshui, and Cangzhou are relatively small. The sensitivity index of soil erosion for forestland and grassland were relatively large (Table 6).

Figure 5 Spatial patterns of resistance surface and ecological corridors.

Spatial patterns of (A) soil and water loss sensitivity, (B) night lighting, (C) geological hazard sensitivity index, (D) basic resistance surface, (E) corrected resistance surface, and (F) ecological source level and ecological corridors.

The highest values of the night light index were mainly distributed in the economically developed cities (Fig. 5B), with urbanized land having the largest average night light indices (Table 6). The average night light index values of Tianjin, Langfang, Beijing, and Tangshan were relatively large (Table 5). The average night light of Chengde and Zhangjiakou were relatively small.

At a regional scale, the distribution of the sensitivity index of geological hazards was highly correlated with the macro-geomorphological gradient and the micro-topographic characteristics of the landscape (Fig. 5C). The geological structure has a pattern of NE–SW and NW–SE directions, being highest in the NE and NW. The average geological hazard sensitivity index of Beijing and Chengde was relatively large, while that of Hengshui and Cangzhouwas relatively small (Table 5). The high values of geological hazard sensitivity were mainly distributed inforestland in the mountainous regions (Table 6).

Based on the resistance value of land use in the Beijing–Tianjin–Hebei area determined by land use type and using methods described by Xie et al. (2015), the resistance value of land use in the Beijing–Tianjin–Hebei urban agglomeration and its corresponding landscape resistance value were determined based on the land use type (Fig. 5D). The resistance value of the forest land, wetland, grassland, farmland, unused land, and construction land were 10, 40, 80, 100, 500, 800, respectively. The revised resistance surface was calculated based on the basic resistance surface, night lighting, soil erosion sensitivity index, and geological hazard sensitivity index (Fig. 5E). The revised resistance surface exhibited significant changes within the same category, and had a relatively distinct spatial heterogeneity, which can more accurately characterize the resistance differentiation of biological migration in the Beijing–Tianjin–Hebei urban agglomeration. The average resistance values of Langfang, Tianjin, and Tangshan were relatively large, while those of Chengde and Zhangjiakou were relatively small (Table 5).

Using the Linkage Mapper plug-in, the ecological source and the modified resistance surface data were input into Arcgis 10.2 to obtain a distribution map showing the locations of ecological corridors (Fig. 5F). At the same time, the length and variation of ecological corridors in each ecological city before and after the resistance surface modification are calculated (Table 7).

Table 7 Changes of ecological corridor length before and after resistance surface correction in Beijing–Tianjin–Hebei urban agglomeration.

Cities	Ecological corridor length (km)	Ecological corridor length (km) (Ecological services, human disturbance)	Variation of length of ecological corridor (km)	
Beijing	451.56	340.51	111.05	
Qinhuangdao	194.61	200	5.39	
Chengde	581.49	657.57	76.08	
Zhangjiakou	961.83	996.24	34.41	
Baoding	1,088.64	1,148.05	59.41	
Tianjin	887.56	1,145.76	258.2	
Tangshan	723.53	966.85	243.32	
Langfang	272.36	332.85	60.49	
Shijiazhuang	1,727.98	1,752.52	24.54	
Cangzhou	371.11	507.12	136.01	
Hengshui	260.15	297.73	37.58	
Xingtai	818.62	915.7	97.08	
Handan	171.66	137.99	33.67	

The length of ecological corridors in some cities of the Beijing–Tianjin–Hebei region have changed significantly over time (Table 7). The length of ecological corridors in Tianjin, Tangshan, Cangzhou, and Beijing changed relatively greatly. Among them, Tianjin has experience major changes in night lighting and soil erosion, Tangshan and Beijing are greatly affected by geological hazards and night lighting, and Cangzhou is sensitive to geological hazards and soil erosion. There are 313 ecological corridors in the Beijing–Tianjin–Hebei urban agglomeration, with a total length of 9,399.59 km (Fig. 5F). The Yanshan and Taihang mountains have a relatively high density of ecological corridors, mainly because of the good quality of the habitats, the strongest correlation between various ecological corridors in the region, and the possibility is greater that various species are able to overcome resistance and migrate. The plains area of central and southern Hebei, has a very sparse distribution of ecological corridors. This primarily occurs because the land use types in this area are mainly farmland and urbanized land with relatively poor quality habitat, and the possibility of various species being able to overcome resistance to migration is relatively small.

Spatial pattern of ecological security pattern

Planning of ecological security pattern

The spatial extent of ecological source areas determined based on the ecological services they provide was 68,882.81 km2; 63.16% of the ecological source areas was within the red line of ecological protection. The areas of overlap between the primary, secondary, and tertiary ecological sources and red line ecological protection were 18,580.57 km2, 14,066.25 km2, and 6,723 km2, respectively. The areas where the identified ecological source area overlapped with the existing ecological protection red line areas gradually decreased over time with the reduction of the level of ecological source area, indicating the rationality of the identification of ecological source areas. Figure 5F shows that the corridors between the plains are relatively long, the ecological source base is basically the third-level ecological source, and the area of ecological source is less. Therefore, it is suggested to add radiation sources on the basis of the original ecological source. The primary ecological source area is arcuate along Yanshan and Taihang mountains. Attention should be paid to the protection of habitats in the aforementioned areas due to their large size and the short ecological corridor. The secondary ecological sources are mainly distributed in the northern mountains of Heibei Province, here, regional habitat restoration should be given more attention, with the goal of forming an ecological security pattern for the Beijing–Tianjin–Hebei urban agglomeration with reasonable spatial distribution and enhanced ecological service functions.

Contrast with the existing ecological security pattern

As noted above, the red line of ecological protection refers to China’s strict land use control boundary demarcated by law for important ecological functional and sensitive areas and vulnerable areas, providing the last line of defense for national and regional ecological security. By geometric correction of the spatial pattern of this ecological protection red line, the delineation of this red line in the analyzed urban area was obtained by vectorization (Fig. 6A). In the Beijing–Tianjin–Hebei urban agglomeration the ecological protection red line area covers 62,329.78 km2, with a total length of the ecological red line corridor of 2,670.57 km. Because the ecological red line corridors are relatively small, this study defines expressways, rivers, and the ecological red line corridor as the current corridor. The habitat in this vector includes forestland, grassland, and wetland in the ecological red line area; it includes nature reserves, water conservation areas, large-scale ecological land use areas and is based on land use data in the Beijing–Tianjin–Hebei planning outline that were regionally vectorized and superimposed. The ecological source area within the study area was extracted and found to cover 4,689.72 km2. Combined with the revised resistance surface, an ecological corridor with human interference factors was generated (Figs. 7A, 7D, 7G, and 7J), and the ecological corridor considering the ecological service separately was generated by combining the basic resistance surface as shown in Figs. 7B, 7E, 7H, and 7K; the ecological corridor considering the ecological service and human interference factors synthetically was generated by combining the modified resistance surface, as shown in Figs. 7C, 7F, 7I, and 7L, respectively. The length of the three ecological corridors and their overlap with the ecological red line corridor and the current corridors are shown in Table 8.

Figure 6 Comparison of ecological security patterns.

(A) The present spatial ecological security pattern and (B) the comprehensive ecological security pattern of the Beijing–Tianjin–Hebei urban agglomeration.

Figure 7 Comparison of ecological corridors.

(A) spatial pattern of human disturbance and ecological red line corridors; (B) ecological service and ecological red line corridors; (C) human disturbance and ecological services and ecological red line corridors; (D) human disturbance and river; (E) ecological service and river; (F) human disturbance and ecological services and river; (G) human disturbance and expressway; (H) ecological service and expressway; (I) human disturbance and ecological services and expressway. Mapping of how the current corridor of the Beijing–Tianjin–Hebei urban agglomeration coincides with: (J) human disturbance corridor; (K) ecological services corridor; (L) human disturbance and ecological services corridor.

Table 8 Comparison of ecological corridor length.

	Ecological corridor (km)	Ecological red corridor overlap (km)	Current corridor overlap (km)	
Ecological service	8,515.68	1,390.06	4,680.62	
Human disturbance	10,626.34	1,545.83	4,751.52	
Ecological service and human disturbance	9,399.59	1,538.76	5,458.98	

The distribution of ecological corridors was generally consistent when the three methods of delineating corridors were compared (Fig. 6B; Table 8). Because of differences in human disturbance and ecological services available in some areas, in this study, the overlapping lengths of the ecological corridors identified by ecosystem services and human interference factors and the current corridor became significantly longer, indicating that comprehensive consideration of ecological services and human interference factors can improve the identification of ecological corridors in urban agglomerations.

Discussion

Determination of ecological sources and level of ecological sources

In terms of ecological source identification special protection areas of the Natura 2000 network that contain forests and agroforest mosaics were selected as core areas to be connected through ecological corridors at a regional scale in the Basque Country (Gurrutxaga, Lozano & Barrio, 2010). This method of directly selecting nature reserves as the core ecological source focuses on considering the functional properties of ecological land patches, but does not consider the spatial structure importance of the ecological source in the whole landscape and the relationship with the surrounding environment. This research effort main goal was to quantitatively evaluate the functions of ecological services, water production services, soil and water conservation services, and leisure services on the basis of absorbing the methods of considering the importance of ecological services in the past, taking into account the importance of the ecological source itself and its structure in the landscape pattern. In the selection of ecological service factors, Zhang et al. (2017) selected biodiversity service, soil conservation service, and water resource security combining the natural environment characteristics of Beijing–Tianjin–Hebei urban agglomeration. On this basis, this paper combines the research results of Peng et al. (2018d) and the ecological protection red line technical guidelines delineated by the Ministry of Environmental Protection of China. Thereby enriching the ecological service supply of the Beijing–Tianjin–Hebei urban agglomeration. In terms of ecological source identification, Zhang et al. (2017) proposed a new evaluation framework integrating ecosystem services importance assessment and landscape connectivity analysis with human ecological demand importance assessment to identify ecological sources. This paper introduces logical coding to establish the spatial relationship between the ecological source and its corresponding ecological services, and describes the spatial pattern of the two by spatial rasterization. Based on logic coding, the ecological service logic map and the ecological source map of importance are generated. From the ecological service logic map, what ecological services are available in the ecological source area can be judged by the location of “1,” “0” and its “1” and “0.” From the ecological source map of importance, the ecological source level can be derived. This is of great significance for the management of refined ecological sources.

Selection of human interference factors

In recent years, the modification of a resistance surface that considers specific ecological problems for specific regions has become an important area of research in the optimization of ecological security patterns. For example, Zhang et al. (2017) used night light data to modify the ecological resistance surface in the Beijing–Tianjin–Hebei urban agglomeration. Peng et al. (2018d) used an imperious index to modify the ecological resistance surface in Shenzhen, China. A surface wetting index was used to modify the ecological resistance surface for pastoral farming areas in semi-arid regions of China (Peng et al., 2018a). Many scholars studied the influence of spatial differences caused by specific ecological problems in different regions on species migration; these factors are collectively referred to as human interference factors, which are used to express the degree of impact that human activities cause to natural environments. For small regions, the effects of a single factor on species migration can be considered for specific ecological environment problems. However, an analysis of the relatively large ecosystem of the Beijing–Tianjin–Hebei urban agglomeration needs to take into account the protection of biodiversity, the restoration of degraded ecosystems, and the sustainable development of the social economy. The purpose of the present study is to systematically help land managers solve regional environmental problems. The singular problems of environmental pollution or biological resources protection have been extended to the systematic analysis and comprehensive study of regional ecological environment problems (Franklin, 1993). Borders of mountains, plains and plateaus can be seen from Fig. 4B. The plateau is located in a small part of the northwest direction of the study area. The plain is located in most of the southwest direction, and the middle part of the plain and plateau is the mountain. The human interference factors selected in this paper include soil erosion sensitivity, geological hazard sensitivity, and nighttime light index. It can be seen from the results in Figs. 5A–5C that the areas of this high geological hazard while soil erosion sensitivity are concentrated in the mountainous areas of urban agglomeration, and the areas of high night light are concentrated in the plains area. The established resistance surface considers the human activity situation of the entire area of Beijing–Tianjin–Hebei. So, the results highlight that the selected human interference factor is relatively reasonable.

Conclusion

Based on ecosystem services and human disturbance factors, this study defined the ecological security pattern of the Beijing–Tianjin–Hebei urban agglomeration, which provides a new holistic framework. The main content of this study include the use of ecosystem services to determine ecological sources, and the establishment of the logical relationship between ecological sources and ecosystem services. Additionally, ecological corridors were identified using a MCR model based on sources and resistance surface modified through human disturbance factors. The results show that water production, soil and water conservation, and carbon fixation sources were mainly distributed in mountainous regions of the study area, recreational sources were mostly distributed in the plains, and the extracted ecological sources improve the recognition of ecological corridors. The modification of resistance surface makes the length of ecological corridors in Tianjin, Tangshan, Cangzhou, and Beijing change significantly, while this modified resistance surface improves the recognition of ecological corridors. Based on the importance of the divided ecological sources and the identified ecological corridors, this study proposes some planning suggestions for the existing ecological security pattern with the goal of forming a reasonable spatial layout and improving the ecological service function of the ecological security pattern of this agglomeration.

Future research of the proposed methodology contains three aspects: (a) a more analytical ecological source identification is that comprehensive consideration of the trade-off and synergy of ecological services to identify the level of ecological source. (b) In terms of resistance surface correction, the nighttime light index is used to directly correct the resistance surface based on land use, the soil erosion sensitivity index is used to correct the resistance surface based on land use, and the resistance surface based on land use is corrected by geological disaster sensitivity. The length of the ecological corridor and the coincidence length of the actual ecological corridor, explore the degree of influence of different human interference factors on the resistance surface. (c) In the aspect of ecological corridor extraction, the identified ecological source level is introduced into the gravity model, and the traditional gravity model is improved to make the identified ecological corridor more scientific and reasonable.

Supplemental Information

Supplemental Information 1 Spatial pattern of potential evapotranspiration in 2015.

The potential evapotranspiration range is 458.99-1327.61 mm, and the color from black to white indicates the grid value from low to high.

Click here for additional data file.

Supplemental Information 2 2015 Precipitation.

The range of rainfall is 320.97-683.342 mm, and the color from black to white indicates the grid value from low to high.

Click here for additional data file.

Supplemental Information 3 Beijing-Tianjin-Hebei border.

The Beijing-Tianjin-Hebei border consists of two parts: 1) the overall boundary and 2) the boundary of 13 cities.

Click here for additional data file.

Supplemental Information 4 Carbon fixation service.

The carbon fixation service is a raster layer and the color from black to white indicates that the raster value is low to high.

Click here for additional data file.

Supplemental Information 5 Spatial pattern of city location.

City location is a point layer. The city location in this article indicates the spatial location of 13 cities in the Beijing-Tianjin-Hebei region. The location of the city is derived from the geometric center of the polygon layers of 13 cities. The polygon layers of 13 cities are derived from Beijing-Tianjin-Hebei border which is another data set provided in this article.

Click here for additional data file.

Supplemental Information 6 Spatial pattern of earthquake acceleration.

Earthquake acceleration is a polygon layer. Different boundaries in the vector file represent different ground motion peaks. The ground motion peak range of the study area in this paper is 0.05-0.3. And it is one of the indispensable data in the calculation of geological hazard sensitivity.

Click here for additional data file.

Supplemental Information 7 Spatial pattern of ecological source.

Ecological source is a polygon layer. This paper includes 150 ecological sources, which are derived from the extremely important areas of vectorized water production services, soil and water conservation services, carbon sequestration services and leisure services.

Click here for additional data file.

Supplemental Information 8 Spatial pattern of engineering rock group.

Engineering rock group is a polygon layer. The attributes of the data mainly include water-rich, water-rich, water-rich index and lithology. And it is one of the indispensable data in the calculation of geological hazard sensitivity.

Click here for additional data file.

Supplemental Information 9 Spatial pattern of expressway.

Expressway is a line layer. This data is obtained by vectorizing the national roads in the Beijing-Tianjin-Hebei 1:10 million basic planning data. A total of 87 line features are included in these line layers.

Click here for additional data file.

Supplemental Information 10 Spatial pattern of fault zone.

The fault zone is the line layer, there are 220 fault zones in this paper. Different fault zones have different linear properties, including quaternary and bedrock zones. And it is one of the indispensable data in the calculation of geological hazard sensitivity.

Click here for additional data file.

Supplemental Information 11 Spatial pattern of geological hazard sensitivity.

Geological hazard sensitivity is raster data, The sensitivity index of geological hazards used nine indicators to calculate the total distribution weight of the layers. The weight of each index was determined by an analytic hierarchy process. Nine indicators includes density of geological hazards, euclidean distance of faults, earthquake acceleration, slope, waviness, slope position, engineering rock group, normalized difference vegetation index and precipitation. Geological hazard sensitivity is calculated through the Raster Calculator module in GIS.

Click here for additional data file.

Supplemental Information 12 Spatial pattern of geological disaster point.

Geological disaster point is a point layer. This paper has identified 6167 geological disasters. The identified geological hazards include various types of disasters such as landslides, mudslides and ground collapses.

Click here for additional data file.

Supplemental Information 13 Spatial pattern of land use.

The year of land use is 2015, and its spatial resolution is 30m. Data were obtained by visual interpretation of Landsat8 OLI imagery.

Click here for additional data file.

Supplemental Information 14 Spatial pattern of leisure and recreation service.

The leisure and recreation service mainly considers the wetlands of the Beijing-Tianjin-Hebei urban agglomeration. Buffer zones of pedestrian walking distance to these wetlands of 5, 10, 15, 20, and 25 minutes (min) were analyzed, which are included in different levels of leisure and recreation.

Click here for additional data file.

Supplemental Information 15 Spatial pattern of slope length.

This is raster data and the slope length factor in the USLE model.

Click here for additional data file.

Supplemental Information 16 Spatial pattern of normalized difference vegetation index.

Normalized difference vegetation index is raster data. The time of Normalized difference vegetation index data is 2015. The data range is from 0-1.

Click here for additional data file.

Supplemental Information 17 Spatial pattern of night lighting.

Night lighting is raster data. In order to facilitate unified calculation, its scope is standardized to 0-1. The highest values of the night light index were mainly distributed in the economically developed cities.

Click here for additional data file.

Supplemental Information 18 Spatial pattern of plant available water content.

Plant available water content is raster data. Each grid corresponds to a GIS raster dataset of water available for plants. Plant available water content refers to the proportion of water supplied to plants in the soil layer. It is a decimal of [0,1]. And it is one of the indispensable data for the InVEST model in the calculation of water conservation.

Click here for additional data file.

Supplemental Information 19 Spatial pattern of rainfall erosivity.

Rainfall erosivity is raster data. Rainfall erosivity is the potential ability of precipitation factors to soil erosion. It is related to rainfall intensity, rainfall duration, rainfall activity, etc. The unit is MJ·mm/(hm^2·h·a). And it is one of the indispensable data for the USLE model in the calculation of soil and water conservation.

Click here for additional data file.

Supplemental Information 20 Spatial pattern of river.

River is a line layer. This data is obtained by vectorizing the river in the Beijing-Tianjin-Hebei 1:10 million basic planning data. A total of 293 line features are included in these line layers.

Click here for additional data file.

Supplemental Information 21 Spatial pattern of the maximum root depth of the vegetation-covered land.

This is implemented based on the GIS-based reclassification function, where the assignment of parameters refers to the Canadell et al’s literature (1996). This is also one of the indispensable datasets for the InVEST model in the calculation of water conservation.

Click here for additional data file.

Supplemental Information 22 Spatial pattern of soil and water conservation service.

Soil and water conservation service is raster data. It is calculated by the USLE model.

Click here for additional data file.

Supplemental Information 23 Spatial pattern of soil and water loss sensitivity.

This is raster data, soil and water loss sensitivity adopts the evaluation indexes of precipitation erosivity, soil erodibility factor K value, slope length and surface vegetation cover, which have been proposed based on the red line technical guidelines for ecological protection.

Click here for additional data file.

Supplemental Information 24 Spatial pattern of soil erodibility factor K value.

This is raster data, and it is one of the indispensable datasets for the USLE model in the calculation of soil and water conservation. The range of data is 0-1.

Click here for additional data file.

Supplemental Information 25 Spatial pattern of modified resistance surface.

This is raster data. The grid resistance surface modified in this paper refers to the land-based resistance surface corrected by geological hazard sensitivity, soil erosion sensitivity index and nighttime light index, which is applied to the minimum cumulative resistance model. Since it is impossible to determine which factors have a greater impact on the resistance surface of an urban agglomeration, the weight of night lighting, soil and water loss sensitivity index and geological hazard sensitivity index is set to 1:1:1.

Click here for additional data file.

Supplemental Information 26 Spatial pattern of water production service.

This is raster data. The amount of water produced refers to the amount of water in the grid unit minus the amount of water after the actual evapotranspiration. It is calculated by the InVEST model.

Click here for additional data file.

Supplemental Information 27 Spatial pattern of red line of ecological protection in Beijing.

The basic ecological red line pattern in Beijing is “two screens and two belts,” the green part of the data is the location of the ecological red line.

Click here for additional data file.

Supplemental Information 28 Spatial pattern of red line of ecological protection in Hebei province.

The basic ecological red line pattern in Hebei Province is “two screens, two belts and multiple points,” the red part of the data is the location of the ecological red line.

Click here for additional data file.

Supplemental Information 29 Spatial pattern of red line of ecological protection in Tianjin.

The basic pattern of the ecological protection red line space in Tianjin is “three zones, one zone and more points,” the red part of the data is the location of the ecological red line.

Click here for additional data file.

Additional Information and Declarations

Competing Interests

Author Contributions

Data Availability

The authors declare that they have no competing interests.

Dongchuan Wang conceived and designed the experiments, performed the experiments, analyzed the data, contributed reagents/materials/analysis tools, authored or reviewed drafts of the paper, approved the final draft, Dr. Dongchuan Wang polished the language of the paper.

Junhe Chen conceived and designed the experiments, performed the experiments, analyzed the data, contributed reagents/materials/analysis tools, prepared figures and/or tables, authored or reviewed drafts of the paper, approved the final draft, junhe Chen discussed and modified the original manuscript.

Lihui Zhang conceived and designed the experiments, performed the experiments, analyzed the data.

Zhichao Sun analyzed the data, prepared figures and/or tables, authored or reviewed drafts of the paper.

Xiao Wang analyzed the data, prepared figures and/or tables, authored or reviewed drafts of the paper.

Xian Zhang analyzed the data, prepared figures and/or tables, authored or reviewed drafts of the paper.

Wei Zhang analyzed the data, prepared figures and/or tables, authored or reviewed drafts of the paper.

The following information was supplied regarding data availability:

The raw measurements are available in Datasets S1–S29.

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
