# Peer review of "Establishing an ecological security pattern for urban agglomeration, taking ecosystem services and human interference factors into consideration"

_PeerJ, doi:10.7717/peerj.7306_

## Round 0.1 · original submission · Minor Revisions

The manuscript was well received and with improvement as suggested will very likely result in acceptance for publication in PeerJ. Review 1 has provided a thorough scrubbing of the manuscript. The main points, for example, include:

1) The use of English needs to be improved throughout the paper, both to ensure that the content can be understood by reviewers and readers, and to make the paper more professional and of archival journal quality.

2) Could you highlight more explicitly the advantages and limitations of the methodology and indicators used in this research? Could you describe more directly what is differentiated in your methodology and why Ecological Security patterns recognition is improved? (e.g. from Zhang et al. 2017 research).

3) A flow chart of the research processes could be useful in order to be easier for a reader or a reviewer to clarify the steps that are followed during each task, sub-models, sub-processes etc.

4) It is highly important to provide if all of the spatial procedures were implemented under a common spatial scale (cell size) (e.g. raster files cell size is varying from 100x100 meters to 1000x1000 meters).

5) Could you give more details about the sensitivity analysis carried out for example in the Resistance Surface Calculation (if it is needed) and what parameters and/or variables have an important influence on your model and the results?

Please address each point raised by both reviews in a revised word doc with track changes and as desired in a rebuttal letter. We look forward to receiving your revised manuscript.

·

Basic reporting

1. Clear and unambiguous, professional English used throughout. (3/5)

1.1 Language and text format
• The use of English needs to be improved throughout the paper, both to ensure that the content can be understood by reviewers and readers, and to make the paper more professional and of archival journal quality. Some examples where the language could be improved include lines:

Line 12: is one of the hot issues - a topic of conversation
Line 32: fruitful results - beneficial results
Line 212: More and more - Increasingly

• Mind the tabs between sentences or parentheses (Line 89, Line 129-142, Figure 2. etc.)

2. Literature references, sufficient field background/context provided. (3/5)

2.1 Introduction and Literature Review

• Introduction (Line 13): However, ecosystem services and human activities are seldom taken into comprehensive consideration in the assessment of ecological security patterns? Already existing literature and previous research indicate this statement?
• Literature review in general covers the topic. However, considering that some of the methodological tools applied in this research are implemented using Geographic Information Systems (G.I.S), a further description in the Literature Review related to G.I.S and Environmental - Ecological Protection may be added in order to highlight how crucial is the use of G.I.S as a Decision Support Tool for Environmental Management and Natural Reserves preservation.
• Some of the References are in the Chinese language therefore, in some parts of the research is not possible to cross the validity of the methods and procedures that are implemented.

3. Professional article structure, figures, tables. Raw data shared. (3/5)

3.3 Figures and Tables

3.3.1 Tables:

Table 5: Weights of selected factors for geological hazard sensitivity
Comments:
• How these weight values are calculated using the Analytical Hierarchy Process – AHP. Some further information may be added including the results from the AHP (e.g. Consistency Ration – C.R and Consistency Index - C.I)
• Is any sensitivity analysis applied for the weighting factors?
Table 7: Logical coding legend may be added (the corresponding raster types).

3.3.2 Figures:

Figures in the Manuscript must be referred to: e.g. Figure 4.A or Figure 1.D, not as B in Figure 4 etc.

Figure 1: Map layout must be corrected (scale bar at the bottom, north arrow top left or right, the legend must be aligned correctly)
Figure 2: Spatial patterns of ecological services. (A) Spatial patterns of water production service. (B) Spatial patterns soil and water conservation service. (C) Spatial patterns of carbon fixation service (D) Spatial patterns of leisure and recreation service.
For Figures, 2.B and 2.D legend classifications are not clearly demonstrated to the maps (colors per each categorization). For example, in Figure 2.D only light blue areas are highlighted.
Figure 3: Non-basic Ecological Line Area must be changed (e.g. light grey) as long as it is not clearly demonstrated to the Legend.
Scalebar can be replaced to the bottom left corner of the Map.
Figure 5: Spatial patterns of resistance surface and ecological corridors (A) Spatial patterns of soil and water loss sensitivity. (B) Spatial patterns of night lighting. (C) Spatial patterns of geological hazard sensitivity index. (D) Spatial patterns of basic resistance surface. (E) Spatial patterns of the corrected resistance surface. (F) Spatial patterns ecological source level and ecological corridors.
Categorization between Figures 5.D (classified intervals) and 5.E (stretched color-bar) is different in spite of the fact that they are demonstrating the same indicator. It must be the same.
Figure 6: Comparison of ecological security patterns. (A) Ecological security spatial pattern (missing gap) (human disturbance) of Beijing−Tianjin-Hebei urban agglomeration. (B) Ecological security pattern (ecological services) of the Beijing-Tianjin- Hebei urban agglomeration. (C) Ecological security pattern (ecological services, human disturbance) of the Beijing-Tianjin-Hebei urban agglomeration. (D) The ecological corridor (human disturbance) of Beijing-Tianjin-Hebei urban agglomeration coincides with the current corridor. (E) The ecological corridor (ecological services) of Beijing-Tianjin-Hebei urban agglomeration coincides with the current corridor. (F) Ecological corridor (ecological services, human disturbance) of the Beijing-Tianjin-Hebei urban agglomeration coincides with the current corridor.
(Comments on Figure titles are not appropriate. This can be referred to the Results section)
Scalebar must be placed to the bottom left corner of the Maps.
Multiple Vector data (Line shapefiles) are presented to the Figures, as a result, it is difficult to clarify all levels of information that are presented.

4. Self-contained with relevant results to hypotheses. (3/5)

• A more detailed explanation for the added value of dividing ecological sources by combining logical coding and ecological service function types combined with the modified resistance surface is needed.

Experimental design

1. Original primary research within the Aims and Scope of the journal. (5/5) (No comment)

2. Research question well defined, relevant & meaningful. It is stated how research fills an identified knowledge gap. (3/5)

• This study will determine the importance of dividing ecological sources by combining logical coding and ecological service function types. Why is this important and how this procedure is differentiated from previous studies? This research question must be further linked with the Litterature Review.
• Why using the soil erosion sensitivity, geological hazard sensitivity, and nighttime light indices, and the modification of ecological resistance surface to identify ecological corridors is important and how this research fills an identified knowledge gap?

3. Rigorous investigation performed to a high technical & ethical standard. (4/5) (No comment)

4. Methods described with sufficient detail & information to replicate. (4/5)

• Section 2.3.1: Ecological service function assessment
Selected indices (water production service, water and soil conservation, carbon fixation service, and leisure and recreation service of the urban agglomeration in combination with the natural environment characteristics) are chosen based on previous work or under the consideration of the authors?
• Section 2.3.2: Ecological Source level
A comparison with previous studies could be useful (e.g. Gurrutxaga et al. 2010: GIS-based approach for incorporating the connectivity of ecological networks into regional planning)
• Section 2.4: Resistance surface
Selected indices (sensitivity to geological hazards, soil erosion, and night light) are chosen based on previous work or under the consideration of the authors?
How these weight values are calculated using the Analytical Hierarchy Process – AHP. Some further information may be added including the results from the AHP (e.g. Consistency Ratio – C.R and Consistency Index - C.I)?
Is any sensitivity analysis applied for the weighting factors?

Validity of the findings

1. Impact and novelty not assessed. Negative/inconclusive results accepted. Meaningful replication encouraged where rationale & benefit to literature is clearly stated. (4/5) (No comment)

2. Data is robust, statistically sound, & controlled. (3/5)

• Section 2.2: Data acquisition and pre-processing. Multiple and non-homogeneous datasets are collected having different spatial analysis and resolution. All available datasets rescaled to a common spatial resolution? Furthermore, no pre-processing procedures are highlighted or mentioned.

To the Supplementary Files:
• Spatial Resolution (Cell Size) is not the same for all .tif images. How the modeling procedures (e.g. Multi-criteria Analysis) are fulfilled with non-homogeneous Spatial Resolution?
• Spatial Reference or Projected Coordinate Systems are not common for all available data. This is not necessarily a problem as long as ‘On the fly’ transformations during processing are possible, it is just mentioned.
• Fault Zones data set has no Spatial Reference.
• Pawc.tif and Geological Hazard Sensitivity.tif seem to be corrupted.

3. Conclusions are well stated, linked to original research question & limited to supporting results. (3/5)

• How this research is differentiated regarding ecological source determination? (e.g. additional indices from previous works, more integrated analysis etc.)
• How modified resistance surface improves the recognition of ecological corridors? Some further explanation may be added.

4. Speculation is welcome but should be identified as such. (4/5)

• Τhe discussion is somehow a repetition of the results. Note that the Discussion section combines the results with current knowledge on the topic and how this research effort highlights some advantages or drawbacks from previous investigations.

Additional comments

This work addresses a topic that I consider important for nature conservation and urban planning in the context of sustainable development and enriches the discussion on this. I congratulate the authors for their extensive data set and their effort to fulfill an integrated analysis related to Ecosystem Services and Nature Conservation and Planning. If there are some weaknesses, they are focused mainly in the use of English, some additional literature review and linkage of the research questions with the results discussion and corrections to some of the Figures and Tables (commented above) which should be improved upon before acceptance.

• The use of English needs to be improved throughout the paper, both to ensure that the content can be understood by reviewers and readers, and to make the paper more professional and of archival journal quality.
• Could you highlight more explicitly the advantages and limitations of the methodology and indicators used in this research? Could you describe more directly what is differentiated in your methodology and why Ecological Security patterns recognition is improved? (e.g. from Zhang et al. 2017 research)
• A flow chart of the research processes could be useful in order to be easier for a reader or a reviewer to clarify the steps that are followed during each task, sub-models, sub-processes etc.
• It is highly important to provide if all of the spatial procedures were implemented under a common spatial scale (cell size) (e.g. raster files cell size is varying from 100x100 meters to 1000x1000 meters).
• Could you give more details about the sensitivity analysis carried out for example in the Resistance Surface Calculation (if it is needed) and what parameters and/or variables have an important influence on your model and the results?

Reviewer 2 ·

Basic reporting

This paper written by Wang et al. entailed an urban agglomeration ecological security pattern taking ecosystem services and human interference factors into consideration. The authors’ investigations indicated a research framework for identifying the ecological security pattern of urban agglomeration and provides scientific guidance for ecological protection and urban planning of the Beijing−Tianjin−Hebei urban agglomeration. In the whole manuscript, in my opinion, it has general scientific hypothesis, complete experimental design, some interesting findings and written well. The adopted methodology is appropriate and results are clear and interpreted well. And, the professional English is also good.

Experimental design

This study uses ecological services to determine ecological sources, divides the importance of ecological sources based on logical coding and functional types of ecological services, combines regional characteristics, synthetically chooses and quantitatively calculates three human disturbance factors and modifies the basic resistance surface of land use that identifies ecological corridor by combining these three disturbance factors. In my opinion, the original primary research is within scope of the PeerJ journal. The experimental design is relatively accurate and reasonable.

Validity of the findings

This study also got some new findings as the ecological source through quantitative evaluation of water production services, soil and water conservation, carbon fixation services, leisure and recreation services. Also, this study determined the importance of dividing ecological sources by combining logical coding and ecological service function types.

Annotated reviews are not available for download in order to protect the identity of reviewers who chose to remain anonymous.

---

## Round 0.2 · Minor Revisions

Thank you for the careful responses to reviewer comments. Please note the remaining minor revisions suggested by reviewer 1. Please consider the revisions and clearly indicate your response in the tracked change document. Note that reviewer 2 provided an annotated pdf.

·

Basic reporting

1. Clear and unambiguous, professional English used throughout. (3/5)

The use of English needs to be further improved throughout the paper as long as some spelling and grammar mistakes exist. Minor changes are provided to the annotated version of the manuscript.

2. Literature references, sufficient field background/context provided. (4/5)

No comment

3. Professional article structure, figures, tables. Raw data shared. (4/5)

Minor changes to some Tables and Figures titles.

4. Self-contained with relevant results to hypotheses. (4/5)

No comment

Experimental design

1. Original primary research within the Aims and Scope of the journal. (5/5)

No comment

2. Research question well defined, relevant & meaningful. It is stated how research fills an identified knowledge gap. (4/5)

No comment

3. Rigorous investigation performed to a high technical & ethical standard. (5/5)

No comment

4. Methods described with sufficient detail & information to replicate. (5/5)

Validity of the findings

1. Impact and novelty not assessed. Negative/inconclusive results accepted. Meaningful replication encouraged where rationale & benefit to literature is clearly stated. (4/5)

No comment

2. Data is robust, statistically sound, & controlled. (4/5)

No comment

3. Conclusions are well stated, linked to original research question & limited to supporting results. (4/5)

No comment

4. Speculation is welcome but should be identified as such. (4/5)

No comment

Additional comments

I want to congratulate and encourage the authors for their extensive and analytical revision. If there are some last minor revisions, they are mainly focusing on some spelling and grammar mistakes which should be improved upon before final acceptance.

Reviewer 2 ·

Basic reporting

The article meets the PeerJ criteria and should be accepted as is.

Experimental design

The article meets the PeerJ criteria and should be accepted as is.

Validity of the findings

The article meets the PeerJ criteria and should be accepted as is.

Additional comments

The article meets the PeerJ criteria and should be accepted as is.

---

## Round 0.3 · accepted · Accept

Thank you for the careful attention to grammatical details noted by reviewer #1. Your article is now acceptable for publication in PeerJ.